# Transporters at the Interface between Cytosolic and Mitochondrial Amino Acid Metabolism

**DOI:** 10.3390/metabo11020112

**Published:** 2021-02-16

**Authors:** Keeley G. Hewton, Amritpal S. Johal, Seth J. Parker

**Affiliations:** 1Department of Biochemistry & Molecular Biology, University of British Columbia, Vancouver, BC V6T 1Z4, Canada; hewt8440@student.ubc.ca (K.G.H.); amritpal.johal@alumni.ubc.ca (A.S.J.); 2British Columbia Children’s Hospital Research Institute, Vancouver, BC V6H 0B3, Canada

**Keywords:** amino acids, transporters, solute carriers, mitochondria, compartmentalization, metabolomics, cytosol

## Abstract

Mitochondria are central organelles that coordinate a vast array of metabolic and biologic functions important for cellular health. Amino acids are intricately linked to the bioenergetic, biosynthetic, and homeostatic function of the mitochondrion and require specific transporters to facilitate their import, export, and exchange across the inner mitochondrial membrane. Here we review key cellular metabolic outputs of eukaryotic mitochondrial amino acid metabolism and discuss both known and unknown transporters involved. Furthermore, we discuss how utilization of compartmentalized amino acid metabolism functions in disease and physiological contexts. We examine how improved methods to study mitochondrial metabolism, define organelle metabolite composition, and visualize cellular gradients allow for a more comprehensive understanding of how transporters facilitate compartmentalized metabolism.

## 1. Introduction

The compartmentalization of metabolic pathways into one or more subcellular organelles is, except for rare cases, a fundamental characteristic of eukaryotic organisms. Metabolic compartmentalization allows for specific pools of enzymes, substrates, and cofactors to be maintained within each organelle, providing unique subcellular conditions to fulfill specialized biochemical functions. The mitochondrion is one such organelle, originating as symbiotic α-proteobacteria that co-evolved within a proto-eukaryote host [1,2]. Many changes to proto-mitochondrial functions have evolved since the initial endosymbiosis occurred [3], complicating our understanding of the metabolic reasoning behind the symbiotic relationship; however, present day mitochondria are complex organelles that participate in broad and critical cellular and biochemical roles. Mitochondria, in addition to canonical ATP generation, play an important biosynthetic role; and amino acid metabolism is intricately linked to this functional output. Notably, amino acids do not freely diffuse across the inner mitochondrial membrane and require specific transport proteins to facilitate their exchange. 

Here we review how plasma membrane and mitochondrial transporters act as important interfaces between compartmentalized metabolic pathways. Further, we discuss key metabolic inputs and outputs of amino acid metabolism that involve mammalian mitochondria with a specific focus on how activity of these pathways is regulated and dysregulated in human disease.

## 2. Compartmentalized Amino Acid Metabolism

### 2.1. Glutamine

Under physiological conditions, glutamine is one of the most abundant amino acids in circulation [4,5]. Glutamine supply is derived from both dietary sources and de novo synthesis, the latter of which requires glutamate and ammonia and is catalyzed by glutamine synthetase (GS) in the cytosol. Activity of GS is well described in the brain as a means of removing excess ammonia by astrocytes, and dysfunctional ammonia metabolism can lead to hepatic encephalopathy and cerebral edema [6,7]. In cells with high proliferative rates (e.g., cancer cells, activated T lymphocytes), glutamine demand outweighs supply and environmental access becomes “conditionally essential” [8,9,10,11,12]. When nutrients become locally limited, several cancers, including pancreatic, glioblastoma, and ovarian, hijack stromal glutamine synthesis as an alternative supply line to fulfill their increased demands [13,14,15]. Furthermore, glutamine deprivation suppresses expansion of activated T lymphocytes, and competition for nutrients within tissues may affect immune responses to pathological states that exhibit hallmark increases in nutrient consumption (e.g., viral infection, cancer) [10,16,17,18]. 

The abundance of glutamine in circulation reflects its robust versatility to satisfy metabolic requirements beyond protein translation (Figure 1). Glutaminolysis represents one of the major catabolic pathways important for the generation of TCA intermediaries, fatty acids, reducing equivalents necessary for oxidative phosphorylation, and non-essential amino acids. The first step of glutaminolysis occurs in the mitochondria and is catalyzed by the enzyme glutaminase (GLS), which converts glutamine to glutamate. There are two isozymes of GLS, the kidney-type (GLS1) and the liver-type (GLS2) [19]. Mitochondrially-produced glutamate serves a number of direct and indirect metabolic roles. For example, glutamate can be oxidized by the NAD(P)^+^-dependent enzyme glutamate dehydrogenase (GLUD1) or contribute its amino-nitrogen for non-essential amino acid synthesis by cytosolic and/or mitochondrial transaminases (e.g., GOT1/2 for aspartate, PSAT1 for serine, GPT1/2 for alanine). Glutamate is also required for glutathione (GSH) synthesis and is used as a backbone for proline and arginine biosynthesis. On the other hand, catabolism of other amino acids (e.g., proline) has been shown in several contexts to be an important source of glutamate. To provide metabolic flexibility in nutrient-limited conditions, pancreatic cancer cells scavenge collagen peptides from the extracellular matrix and utilize proline as an anaplerotic source when glutamine levels are low [20]. Furthermore, proline catabolism by proline dehydrogenase (PRODH) has also been shown to be an important source of glutamate in metastasizing breast cancer cells [21]. In other contexts, mitochondrial pyrroline 5-carboxylate reductase 1 (PYCR1) redirects excess mitochondrial NADH and/or glutamate towards proline synthesis in isocitrate dehydrogenase 1 (IDH1) mutant glioma cells, leading to a partially uncoupled TCA cycle that allows cells to regulate mitochondrial NAD^+^/NADH [22]. 

While the capability to utilize glutaminolysis pathways is transcriptionally inherent in the genome of all cells, the tendency to shunt glutamine towards them is likely tissue-specific, cell-type specific, and depends on conditional metabolic needs [23]. For example, neonatal mammals utilize proline, not glutamine, as the major anabolic input for arginine synthesis [24]. On the other hand, a shift to glutaminolysis as the major glutamate source is intrinsic across many cancer contexts. Upregulation of GLS by oncogenic pathways is a common mechanism for this metabolic shift. The oncogenic transcription factor c-Myc has been implicated as one driver of GLS1 upregulation by suppressing the inhibitory effects on GLS1 translation by miR-23a/b [25]. This upregulation of GLS1 through post-transcriptional mechanisms has also been attributed to other pro-neoplastic factors. NF-κB member p65 downregulates miR-23a transcription in leukemic cells, while HSF1, which is ubiquitously expressed in several cancer types, suppressed transcription of the GLS1-inhibitor miR-137 [26,27]. Additionally, the transcription factor c-Jun, a downstream effector of oncogenic-Dbl and the JNK-MAP kinase pathway, was shown to directly bind the GLS1 promoter and promote its upregulation [28]. In contrast to GLS1, GLS2 has been identified as a target for the tumor suppressor p53 and thus may support anti-cancer properties [29,30]. This apparent contradiction may be a result of differences in properties between the two GLS isozymes and requires further investigation.

Other oncogenes have been further shown to mediate pro-cancer effects through upregulation of glutaminolysis. In pancreatic ductal adenocarcinoma cells (PDAC), oncogenic KRAS was shown to shift glutamine metabolism by upregulating cytoplasmic GOT1 and downregulating GLUD1, stimulating a pathway in which glutamine-derived aspartate from the mitochondria is used as a metabolite to generate cytosolic OAA [11]. Subsequent activity of malate dehydrogenase (MDH) and cytosolic malic enzyme (ME1) supply PDAC cells with reduced pyridine nucleotides necessary for redox homeostasis [11]. In a separate study, PDAC cells cultured in acidic conditions also exhibited an increased dependence on glutamine for redox homeostasis and anaplerosis [31]. In colorectal cancer (CRC), mutations in the PIK3CA gene, which encodes for the p100α subunit of PI3K, lead to upregulation of GPT2 and reliance on glutamine-derived TCA intermediates to sustain growth [32]. Additionally, the liver receptor homolog 1 (LRH1) has been implicated as a transcription factor, which drives tumor formation via effects on glutamine metabolism in hepatocellular carcinoma (HCC) [33]. Overall, the upregulation of glutaminolysis in cancer cells is near ubiquitous and achieved through many different oncogenic effectors.

Inhibiting aberrant glutaminolysis has largely focused on targeting mitochondrial glutaminase activity. BPTES and the more soluble and bio-available CB-839 selectively inhibit GLS1 and have been investigated as anti-neoplastic agents in several contexts [34,35,36,37,38]. However, certain cancer types (e.g., pancreatic, lung) demonstrate contradicting sensitivity to GLS inhibition in vitro and in vivo, suggesting that tumors may be more glutaminolysis independent in vivo than modeled in culture [39,40]. Further, these studies highlight the plasticity of glutamine and glutamate metabolism and suggest that cells may autonomously re-route metabolic flux to supply glutamate through other means (Figure 1). Glutamate is produced during the synthesis of purine and pyrimidine nucleobases and the glycosylation subunit N-acetyl-glucosamine (GlcNAc). Purine and pyrimidine synthesis utilize the γ-nitrogen of glutamine to generate 5-phospho-β-d-ribosylamine (PRA) and carbamoyl phosphate (CP) by phosphoribosyl pyrophosphate amidotransferase (PPAT) and the carbamoyl phosphate synthetase (CPS) domain of the CAD complex, respectively [41,42,43]. Cytosolic glutamine is also a substrate for glutamine-fructose 6-phosphate aminotransferase (GFPT1), which is used to produce glutamate and glucosamine 6-phosphate (GlcN6P) a precursor for O-linked N-acetylglucosaminylation [44]. Furthermore, cytosolic asparagine synthesis by asparagine synthetase (ASNS) yields glutamate as well. Given the numerous glutamate supply routes, efforts to target the glutamine demands in cancer have broadened to identify antagonists targeting more than one glutamine-dependent enzyme simultaneously. 6-diazo-5-oxo-L-norleucine (DON) was developed decades ago as a potential anti-neoplastic agent for its inhibitory activity against many glutamine-dependent enzymes, including glutaminase and glutamine amidotransferases [45]. However, gastrointestinal (GI) toxicity in the majority of patients receiving DON limited its clinical use [46]. More recently, pro-drug forms of DON have been developed with enhanced delivery properties to either the brain or tumors and reduced GI toxicity, which has reinvigorated interest in using glutamine antagonists as antitumor agents [47,48,49,50,51]. An alternative strategy is limiting cancer cell access to glutamine by inhibiting transporter-dependent uptake. Two of the most well documented glutamine transporters are from the SLC1A and SLC38A solute carrier (SLC) families, and the more promiscuous Na^+^/Cl^−^-dependent SLC6A14/ATB^0,+^ transporter can also play a role in importing glutamine [52,53,54]. SLC1A5/ASCT2 inhibitors have been identified and exhibit promising anti-tumor properties in preclinical models [55,56,57,58,59]. Furthermore, targeting secondary glutamine transporters (e.g., SLC38A2, SLC6A14) genetically or pharmacologically (e.g., α-methyltryptophan) significantly suppresses amino acid homeostasis and tumor growth in pancreatic cancer [60,61].

### 2.2. Aspartate

Aspartate is an acidic non-essential amino acid that can be acquired by either de novo synthesis and/or import from external sources. However, circulating levels of aspartate in physiological conditions are low (~10 µM) and maintained by liver aspartate transaminases; thus, synthesis likely provides the majority of cellular aspartate in most contexts [4]. Biosynthesis of aspartate is carried out via aspartate aminotransferases (glutamic-oxaloacetic transaminases) in the cytosol (GOT1) and in the mitochondrial matrix (GOT2), which as discussed above utilize glutamate as the amino-nitrogen source. Aspartate has many biosynthetic fates within the cell (e.g., proteins, nucleotides, and amino acids) and also serves as an exchange factor for the aspartate-glutamate carrier (AGC1/AGC2), an essential component of the malate-aspartate-shuttle (MAS) (Figure 2). MAS is responsible for transferring electrons from cytosolic NADH to mitochondrial NADH, as reducing equivalents (e.g., NAD(P)H) cannot directly cross the inner mitochondrial membrane. However, recent studies identified that SLC25A51 and SLC25A52 facilitate mitochondrial NAD^+^ transport [62,63,64]. Subsequent activity of the MAS and/or UCP2 is required to export aspartate into the cytosol where it can be used as a proteinogenic source and/or a precursor for arginine and asparagine synthesis [65,66].

In many contexts, aspartate is predominantly synthesized by mitochondrial GOT2 and is suggested to be one output of mitochondrial electron transport chain (ETC) activity in proliferating cells [11,67,68,69,70,71,72]. Although ATP is another major output of ETC activity, proliferating cells with sufficient access to glucose can switch to aerobic glycolysis to largely satisfy these requirements [67]. Aspartate serves a biosynthetic role, acting as a nitrogen donor for adenine synthesis and a carbon backbone via orotate for pyrimidine synthesis. The availability of aspartate has been suggested to be limiting for the proliferation of certain cancers. Sullivan et al. utilized a guinea pig asparaginase (gpASNase1) to supply tumors with asparagine-derived aspartate and observed enhanced tumor growth in HCT116 and AL1376 colorectal and murine PDAC cell lines, respectively [73]. Interestingly, gpASNas1 had little to no effect on the human AsPC1 tumor growth. Similarly, some cancer cells utilize a plasma membrane glutamate and aspartate transporter, SLC1A3, to provide aspartate in conditions where de novo synthesis is restricted by ETC inhibition [74]. Environmental acquisition of aspartate by SLC1A3 has also been implicated in hypoxic microenvironments or in response to glutamine restriction [72,74,75]. Hypoxia reportedly suppresses mitochondrial aspartate biosynthesis via HIF1α-dependent down-regulation of GOT1 and GOT2 in Von Hippel-Lindau (VHL)-deficient renal carcinoma cells [76]. However, pancreatic cancer cells have been shown to sustain aspartate biosynthetic fluxes in oxygen tensions as low as 0.1% O_2_ through activity of Complex III+IV containing respiratory supercomplexes, which are suggested to promote efficient respiration in limiting oxygen environments [77]. Notably, maximal HIF stabilization occurs in ~1% O_2_, well above tensions where oxygen becomes limiting for mitochondrial respiration [78]. Although glutaminolysis provides cells with the majority of carbon necessary to synthesize aspartate, in cancer subtypes driven by TCA cycle deficiencies (e.g., SDH- or FH- deficiency), pyruvate carboxylase activity can divert glucose-derived pyruvate to supply oxaloacetate necessary for this anabolic function [79,80,81,82,83,84]. This shift to PC-dependent aspartate synthesis was also observed in PDAC tumors in vivo and in breast and lung cancer cell lines exposed to hypoxic oxygen tensions [77,85]. Taken together, aspartate is a critical anabolic metabolite necessary to supply nucleotides for proliferating cancer cells; however, its synthesis from glutamine- and/or glucose-derived pathways are complex and highly dependent on the environmental context and nutrient availability. 

Cytosolic aspartate is utilized by ASNS and ASS1 for asparagine and arginine biosynthesis, respectively (Figure 2). The production of these amino acids supports protein translation, but also play indirect roles for proliferation. For example, asparagine acts as an amino acid exchange factor to facilitate the influx of other amino acids (e.g., serine, threonine) necessary to regulate mammalian target of rapamycin complex 1 (mTORC1) activity and proliferation [86]. Arginine is a major source of cellular nitric oxide (NO), through activity of iNOS, or is catabolized by arginase as the final step of the urea cycle. Arginine, and other basic amino acids (e.g., lysine, ornithine), can also transport into the mitochondria by SLC25A29 [87]. Expression of SLC25A29 was shown to be elevated in several cancer cell lines and important for NO production by a mitochondrial NOS [88]. Activity of extrahepatic arginase 2 (ARG2) also regulates mitochondrial NO production [87,89,90]. Notably, several cancers down-regulate the activity of these pathways via silencing of ASS1 and/or ASNS expression, creating a dependence (auxotrophy) for environmental and/or stromal acquisition of these amino acids [91,92,93,94]. Silencing of ASS1 and/or ASNS may provide a selective advantage for cancer cells, allowing for diversion of aspartate towards other anabolic pathways such as nucleotide biosynthesis. Importantly, activity of the MAS and/or other mitochondrial aspartate transporters (e.g., UCP2) represent a key step for regulating compartmental availability of this critical amino acid [65,66].

### 2.3. Serine, Glycine and Alanine

The metabolism of small neutral amino acids serine, glycine, and alanine occurs in both the cytosol and mitochondria and has implications for physiology and human diseases (Figure 3). Serine consists of a simple hydroxymethyl side chain and is either taken up or synthesized de novo from the glycolytic intermediate 3-phosphoglycerate by three cytosolic enzymes, phosphoglycerate dehydrogenase (PHGDH), phosphoserine aminotransferase (PSAT1), and phosphoserine phosphatase (PSPH). Activity of serine synthesis is important for normal development, as deletion of *Phgdh* causes embryonic lethality in part due to neurological defects [95]. Serine, specifically D-serine, is thought to be a critical excitatory neurotransmitter acting as a co-agonist of the N-methyl D-aspartate (NMDA) receptor on neurons, and depletion due to deficient synthesis or racemase activity likely leads to catastrophic neurotoxicity [96]. Abnormal D-serine levels in the brain are thought to contribute to neurodegenerative disorders such as Alzheimer’s disease and schizophrenia [97,98,99,100]. Expression of serine biosynthesis enzymes are highly regulated by several factors, including NRF2-ATF4 [101,102], c-Myc [103], and hypoxia inducible factors [104]. Many of these transcriptional regulators are altered in cancer and contribute to increased serine synthesis flux, but PHGDH expression was also found to be amplified through copy number gain of a genomic region on chromosome 1p12 in a subset of breast and melanoma [105,106]. PHGDH expression has been demonstrated to support tumor growth specifically in low serine environments, such as cerebrospinal fluid where concentrations are significantly lower than plasma, and dietary restriction of serine and glycine reduces tumor growth in preclinical cancer models and enhances activity of mitochondrial inhibitors [107,108,109,110,111]. Furthermore, a subset of PDAC cells downregulate serine synthesis enzymes, and neuronal supply has been shown to supply cancer cells with serine specifically in null environments [112]. As PHGDH is thought to be the rate-limiting step of serine synthesis and important for the proliferation of PHGDH-amplified cancer cell lines, several inhibitors have been developed [113,114,115]. Notably, serine synthesis and uptake can occur in parallel with catabolism, depending on the context and cell intrinsic demand for serine and/or its many catabolic outputs.

Serine is utilized for glycine synthesis as well as ceramide and sphingolipid synthesis, nucleotide synthesis, folate-mediated one carbon metabolism (FOCM), S-adenosyl methionine regeneration for methylation reactions, and transsulfuration for cysteine biosynthesis (Figure 3). Glycine synthesis requires the cytosolic and mitochondrial enzymes serine hydroxymethyltransferase (SHMT1/2) producing one-carbon units in the form of 5,10-methylene-tetrahydrofolate (5,10-meTHF). Subsequent activity of methylenetetrahydrofolate dehydrogenases (MTHFD1/1L/2) releases formate, which can be transported across mitochondrial membranes. The resulting metabolic cycle acts as a shuttle for NAD(P)H reducing equivalents and one-carbon units required for thymidine and purine synthesis. The directionality of the FOCM metabolic cycle operates predominantly oxidatively in the mitochondria but can reverse to maintain one-carbon supply for nucleotide synthesis in cases where mitochondrial isoforms are deleted [116,117,118]. High activity and dependence on SHMT1/2 for proliferation has been demonstrated in multiple cancer contexts, and inhibitors targeting these enzymes have been developed [105,106,119,120]. In replete environments, serine catabolism by SHMT2 can occur in excess and release of glycine and formate, termed “formate overflow”, has been reported for several cancer and non-transformed cell lines and in mice [121]. Through this mechanism, serine catabolism acts as a significant source of both ATP and NAD(P)H, and flux through this pathway was demonstrated to support oxidative mitochondrial metabolism [122]. In response to pharmacological inhibition of respiration expected to increase mitochondrial NADH/NAD^+^, mitochondrial serine catabolism is sustained, whereas other enzymatic NADH sources (e.g., pyruvate dehydrogenase) were feedback inhibited [123]. Thus, serine metabolism is highly complex and can provide cells with glycine, one-carbon units, NAD(P)H, and ATP depending on the context and cellular demand [124].

Glycine can also contribute one-carbon units through mitochondrial activity of the glycine cleavage system (GCS), which releases CO_2_, NH_3_, and 5,10-meTHF (Figure 3). Notably, activity of GCS in many cancer cell lines was found to be low relative to serine catabolism [125]. However, in the context of cancer cell lines with high mitochondrial SHMT2 flux, activity of GCS is necessary to clear excess glycine and prevent a build-up of the toxic byproducts aminoacetone and methylglyoxal derived from the interconversion of glycine and threonine [126]. Methylglyoxal was found to accumulate in non-small cell lung cancers relative to normal tissue, and sequestration of toxic methylglyoxal requires glutathione (GSH) and activity of glyoxalase (GLO1) to prevent cellular damage [127]. Activity of the GCS has been shown to be important for the maintenance of stem cell pluripotency through epigenetic regulation [128]. Extracellular glycine can also be used for SHMT-dependent serine synthesis but requires an exogenous source of formate [125]. In addition to FOCM, glycine is also a precursor required for the synthesis of GSH and δ-aminolevulinic acid (δ-ALA; mitochondrial) necessary for heme biosynthesis. Synthesis of glutathione occurs in the cytosol, requiring mitochondrial export of glycine and GSH import into intracellular organelles including the mitochondria, which contains ~10–15% of cellular GSH at a similar concentration to the cytosol [129]. Maintenance of GSH pools is important for regulating the activity of proteins sensitive to post-translational oxidation of cysteine residues (e.g., PTP1B) [130,131,132]. Insight into the activity and downstream role of serine and glycine metabolism can be gained from examination of extracellular uptake and secretion; however, cytosolic-mitochondrial exchange is equally important and requires a number of plasma membrane and mitochondrial transporters [121,133]. 

Alanine synthesis requires cytosolic and/or mitochondrial glutamic-pyruvic transaminases (GPT1/2). Physiological synthesis occurs in skeletal muscle from pyruvate and glutamate derived from glycolysis and BCAA catabolism, respectively. Alanine secreted by muscles provides the carbon necessary for gluconeogenesis in the liver, which in turn provides glucose back to muscles and sequesters the nitrogen produced from alanine catabolism as urea [134,135,136,137]. The resulting glucose/alanine cycle, referred to as the “Cahill cycle”, is an important organ crosstalk relevant during normal physiology, exercise, fasting, and disease [138]. Dysregulation of this cycle has been proposed to occur in cancer patients, whereby increased protein turnover and/or muscle breakdown (“cachexia”) releases alanine, BCAAs, and other amino acids for hepatic gluconeogenesis [139,140,141]. Elevated hepatic alanine-to-glucose conversion was measured in lung and other cancer patients, but plasma alanine levels remained mostly stable [142,143,144,145]. Hepatocytes express both cytosolic (GPT1) and mitochondrial (GPT2) isoforms required for de novo alanine synthesis and catabolism. However, biochemical parameters and studies suggest that GPT1 (Km, ala = 34 mM) and GPT2 (Km, ala = 2 mM) exhibit preference towards alanine anabolism and catabolism, respectively, although this was highly dependent on the method used to ascertain directionality [146,147,148,149,150]. Recent evidence suggests that pancreatic cancer cells have a high demand for alanine, and scavenge alanine from stromal sources (e.g., activated stellate cells) [60,151]. The majority of human pancreatic cancer cells selectively express GPT2, at both the transcript and protein level, suggesting that alanine metabolism occurs mainly in the mitochondria [60]. Notably, mitochondrial alanine catabolism by GPT2 requires activity of a mitochondrial alanine transporter, which was functionally identified decades ago but remains unknown [147]. Low expression of cytosolic GPT1, which catalyzed alanine synthesis from pyruvate in hepatocytes, and alanine uptake was suggested to provide pancreatic cancer cells with the capacity to retain pyruvate in the cytosol and support aerobic glycolysis [60,152]. In contrast, naïve T lymphocytes require alanine for activation as neither GPT1 nor GPT2 are expressed at sufficient levels [153]. Alanine production from pyruvate was found to be important for the metastasis of breast cancer cells, providing a source of α-ketoglutarate used for collagen hydroxylation and extracellular matrix (ECM) remodeling [154]. Importantly, it has been suggested that transport across the plasma membrane may be the main rate-limiting step of alanine metabolism at extracellular concentrations <1 mM [147,155,156]. Normal plasma levels of alanine are ~0.2–0.4 mM, but elevated levels (~1 mM) have been measured intratumorally, suggesting altered alanine metabolism and availability in cancer and tumor-associated stromal cells [157,158]. Notably, SLC38A2/SNAT2 was identified to be the main concentrative alanine transporter utilized by pancreatic cancer cells and targeting SLC38A2 was sufficient to suppress alanine uptake by pancreatic cancer cells and cause significant re-wiring of compartmentalized pyruvate metabolism [60]. Taken together, these studies suggest that perturbing alanine metabolism in cancer is possible by altering plasma membrane transport, and mitochondrial alanine transport may be a key player in glucose-pyruvate-alanine metabolism by skeletal muscle, hepatocytes, and cancer cells.

### 2.4. Branched-Chain Amino Acids

Branched chain amino acids (BCAAs) include leucine, isoleucine, and valine and are derived from dietary sources. Because of their essentiality in mammals, BCAA transport and sensing in addition to catabolic mechanisms of acquisition (e.g., autophagy, macropinocytosis) has attracted much interest. Cellular uptake of BCAAs is mainly facilitated by the L-type amino acid transporter (SLC7A5/LAT1), which requires dimerization with SLC3A2/CD98 to function. It also transports aromatic amino acids (e.g., tyrosine, phenylalanine) (Figure 4) [159,160]. Notably, LAT1 is sodium-independent and relies on other amino acids to serve as exchange factors to facilitate net BCAA import [159]. Leucine is well characterized to influence mTORC1 signaling, which is aberrantly activated across many cancer types [161]. Leucine can activate mTORC1 signaling through direct sensing by Sestrin2 and disruption of the Sestrin2-Gator2 interaction, triggering a signaling cascade through downstream effectors (e.g., eukaryotic translation initiation factor 4E binding protein 1, p70-S6 kinase, ULK1) [161,162,163,164]. These signals coordinate proliferation through activity of autophagy and protein, lipid, and nucleotide synthesis.

Because of the abundant expression of SLC1A5/ASCT2 and LAT1, it has been suggested that ASCT2-dependent glutamine uptake may serve as the exchange factor for BCAA influx by LAT1. However, ASCT2 is dispensable for the proliferation and mTORC1 signaling in many cancer lines, and ASCT2 functions primarily as an exchanger unable to concentrate glutamine sufficiently to drive LAT1 activity [55,165,166]. Thus, secondary active glutamine transporters (e.g., SNAT1/SLC38A1, SNAT2/SLC38A2, SLC6A14/ATB^0,+^) are more likely to contribute to glutamine concentration for LAT1-mediated exchange. However, deletion of SLC38A2 in pancreatic cancer failed to impact either BCAA or glutamine uptake flux despite significantly decreasing intracellular glutamine levels [60]. Rather, transporter cooperativity between glutamine and BCAA transporters may be more important for level maintenance. Indeed, LAT1 knockout results in a ~90% decrease in leucine transport in hepatocellular carcinoma cells but fails to illicit proliferative defects, and knockdown or inhibition of LAT1 did not negatively impact mTORC1 re-activation following EAA stimulation [165,167]. Furthermore, knockout of SLC3A2/CD98 abolished ~90% of leucine uptake by LAT1 in colon adenocarcinoma cells, but proliferative defects and activation of the GCN2-linked amino acid stress response were not observed [168]. Thus, plasma membrane transport of BCAA and/or glutamine may not be limiting or is highly dependent on the cellular context, and minimal transport capacity may be sufficient to satisfy the biosynthetic and catabolic demands for these amino acids. In contrast, LAT1 was significantly upregulated in an *Apc^fl^*^/*fl*^*; LSL-Kras^G12D^*^/*+*^*; Villin^CreER^* mouse model of colorectal cancer, and targeted deletion of *Slc7a5* resulted in delayed tumorigenesis and improved survival [169]. Furthermore, JPH203, a small molecule inhibitor of LAT1, has shown significant pre-clinical efficacy in colorectal cancer and T-cell lymphoblastic lymphoma/leukemia and was well-tolerated in a Phase I study in patients with advanced solid tumors [170,171,172]. Other transport systems can facilitate BCAA uptake, including the Na+-dependent SLC6A19/B0AT1, which may contribute to differing sensitivity in response to LAT1-deletion [173,174]. Inhibitors targeting SLC6A19/B0AT1 have been developed using in silico and high-throughput screening approaches [175,176].

Aside from being used for protein synthesis, BCAAs can contribute to anabolic and bioenergetic outputs important for human physiology and dysregulated activity is attributed to multiple diseases (reviewed in [161,177,178]) (Figure 4). Through catalytic activity of highly reversible branched chain aminotransferases (BCAT1/2) localized within the cytosol (BCAT1) or mitochondrial matrix (BCAT2), BCAA catabolism provides cells with amino-nitrogen for glutamate synthesis as well as branched chain ketoacids (BCKAs) (e.g., α-ketoisocaproic, KIC; α-ketoisovaleric, KIV; α-keto-β-methylvaleric, KMV) that contribute to acyl-CoA synthesis, lipogenesis, and TCA cycle metabolism. While BCAT2 is ubiquitously expressed, BCAT1 is selectively expressed in the brain, ovary, and placenta [179]. BCAT1 is commonly up-regulated in many different cancer lines, such as human glioblastoma, breast cancer, and non-small cell lung carcinoma (NSCLC), while BCAT2 seems more important for pancreatic cancer [180]. Furthermore, elevated plasma BCAA levels are associated with several diseases, including cardiovascular disease, pancreatic cancer, and breast cancer [139,181,182]. In the mitochondria, BCKAs can undergo irreversible decarboxylation by the branched chain α-ketoacid dehydrogenase (BCKDH) complex, which consists of three subunits (E1, E2, and E3). Activity of BCKDH is negatively regulated by the phosphorylation status of the E1 subunit. BCKDH kinase (BCKDK) and the Mg^2+^/Mn^2+^-dependent 1 K protein phosphatase (PPM1K) coordinate the activity of BCKA oxidation. Activity of PPM1K was shown to positively regulate BCAA catabolism important for leukemogenesis [177,183]. Furthermore, defective BCKA oxidation drives the inborn error of metabolism maple syrup urine disease (MSUD), and dysregulated BCKDH activity is also attributed to several human diseases (e.g., diabetes, cancer) [184]. 

Acyl-CoA products of BCAA oxidation (e.g., acetyl-CoA, propionyl-CoA, succinyl-CoA) have the potential to contribute carbon for oxidative TCA cycle activity and/or lipogenesis, suggesting that BCAA may serve as an important fuel source for proliferative cells. In addition, acetyl-CoA derived from leucine can provide direct proliferative signals through acetylation of Raptor via EP300, which in turn negatively regulates autophagosome formation and activates mTORC1 signaling [185,186]. Whether this represents a major metabolic contribution, particularly to the TCA cycle, depends highly on the context. The metabolic contribution of BCAA-derived acyl-CoA has been extensively characterized in mutant Kras-driven tumors (e.g., pancreatic, lung) given the correlation between elevated plasma levels and disease progression [139]. In acute myeloid leukemia (AML), human pancreatic cancer, and colorectal cancer cells, as well as in *LSL-Kras^G12D^*^/*+*^; *Trp53^flox^*^/*flox*^-driven lung and pancreatic tumors, ^13^C-labeled BCAAs contributed minimally to mitochondrial TCA cycle intermediates irrespective of which BCAT1/2 isoform is expressed in each context [180,187,188,189,190]. In contrast, cancer-associated fibroblasts derived from human pancreatic tumors showed higher BCAA oxidation flux than pancreatic cancer cells, and BCKAs secreted from CAFs were incorporated into the TCA cycle in human pancreatic cancer cells through subsequent oxidation [191]. Similarly, ^13^C-KIC, derived from leucine catabolism, was shown to be oxidized by tumors in a rat glioma model using hyperpolarized nuclear magnetic resonance (NMR) spectroscopy [192]. Transport of BCKAs across the plasma membrane is mainly facilitated by monocarboxylate transporters, MCT1/SLC16A1 and MCT4/SLC16A4, allowing cells to share pools of circulating BCKAs to convert to BCAAs if needed [193,194,195,196]. In adipocytes, BCAAs represent a major anaplerotic and lipogenic source. Acetyl-CoA or propionyl-CoA is utilized for even- or odd-chain fatty acid synthesis and, in addition to succinyl-CoA, contributes significantly to TCA cycle intermediates (e.g., citrate) [197,198,199]. Adipose tissue can also utilize BCAA catabolism to generate mono-methylated branched-chain fatty acids through promiscuous activity of carnitine acetyltransferase (CRAT) and fatty acid synthase (FASN) [200]. Notably, the methylmalonyl-CoA mutase required to convert propionyl-CoA to succinyl-CoA is B12-dependent, and odd-chain fatty acids and methylmalonic acid (MMA) accumulate in adipocytes only when cultured in media deficient in cobalamin (e.g., DMEM) [199]. In a recent study, increased MMA levels in circulation correlate with increasing age, and MMA was found to promote an epithelial-mesenchymal transition (EMT)-like phenotype and contribute to increased tumorigenesis [201].

## 3. Mitochondrial Amino Acid Carriers

Many of the metabolic fates of the above discussed amino acids center in and around the mitochondria, and mitochondrial transporters likely play a critical role in facilitating the activity of amino acid metabolism (Figure 1, Figure 2, Figure 3 and Figure 4). Eukaryotic mitochondria comprise an outer and inner membrane that separate the internal matrix from the cytosol. The two mitochondrial membranes form complex substructures that include cristae and contact sites between membranes and with other organelles, all of which can impact mitochondrial function [202,203,204]. The outer mitochondrial membrane (OMM) is highly permissive up to ~5 kDa, and translocases are employed to import mitochondrial-targeted proteins across both inner and outer membranes. However, the inner mitochondrial membrane (IMM) is impermeable to most small molecules, similar to other cellular membranes, allowing the mitochondrial matrix to maintain a distinct metabolite composition compared to the surrounding cytosol. Specific mitochondrial transporters are required to facilitate exchange of ions and metabolites; such as adenine nucleotides, amino acids, acyl-carnitines, and small organic acids. The 53 membered SLC25 family represents the largest component of mitochondrial transporters. Other transmembrane protein families, such as the sideroflexin family (SFXN), the mitochondrial pyruvate carrier (MPC1/2), certain ATP-binding cassette transporter (ABCB) isoforms, and splice variants of other solute carriers (SLCs), also contribute to mitochondrial transport. Excellent reviews of our current knowledge of mitochondrial transporters can be found elsewhere [205,206,207]. Recent progress has been made on the identification of mitochondrial amino acid carriers, including those that transport serine (SFXN1/3), glutamine (mitochondrial targeted SLC1A5 variant), and branched chain amino acids (SLC25A44) [208,209,210]. 

Despite the important role that serine plays for nucleotide, glycine, and one-carbon metabolism and the compartmentalization of these pathways; the transporter(s) involved in its transport into the mitochondria have only recently been identified. Kory et al. identified that sideroflexin 1 (SFXN1) and other SFXN homologs act as inner mitochondrial membrane-localized serine transporters [209]. To identify the mitochondrial serine transporter, Kory et al. utilized a functional genetic screening approach in cells lacking the cytosolic arm of FOCM, creating an increased reliance on mitochondrial serine transport for proliferation. Functionally, SFXN1 was important for glycine pool maintenance and folate charging, owing to defective oxidative mitochondrial serine-dependent FOCM activity. SFXN1-null cells were not auxotrophic for glycine, suggesting that other sideroflexin homologs, of which there are five, may relay some compensatory activity. Through subsequent functional genetic screening in SFXN1-null cells, the authors found that SFXN3 was a likely candidate for redundant mitochondrial serine transport. In vitro liposome reconstitution of SFXN1 and stable isotope tracing suggest that SFXN1 is capable of importing serine and other small neutral amino acids, including alanine, cysteine, and glycine. This study fills an important gap in our knowledge of mitochondrial serine transport and highlights the power of functional genetic screening, stable-isotope tracing, and metabolomics to characterize transporter function in relevant contexts. Given the redundant function of some sideroflexin homologs, complete suppression of mitochondrial transport may require inhibition of multiple targets to treatment of aberrant serine metabolism in diseases like cancer. Mitochondrial glycine import and/or export may also play an important role in facilitative FOCM and purine and glutathione biosynthesis. SLC25A38, and its yeast homolog Hem25, was recently characterized as a mitochondrial glycine transporter [211]. Mutations in *SLC25A38* give rise to congenital sideroblastic anemia, caused by a defect in heme biosynthesis [212]. Notably, SHMT2 activity could, in theory, provide mitochondrial glycine for heme biosynthesis; however, the authors found that Shm1 and Shm2 (yeast homologs of SHMT1 and SHMT2, respectively) did not significantly contribute to heme synthesis [211]. Notably, it is not clear whether SLC25A38 also facilitates mitochondrial glycine export, which may be important for purine and glutathione synthesis in nutrient-limited environments. 

The anabolic and bioenergetic outputs of glutaminolysis require activity of a mitochondrial glutamine transporter that was known to exist but only recently identified [213]. Yoo et al. identified a variant of the plasma membrane transporter SLC1A5 localized to the mitochondrial inner membrane capable of importing glutamine (SLC1A5_var) [208]. To identify this candidate, the authors hypothesized that a mitochondrial glutamine transporter would share structural homology to its plasma membrane equivalent (pmSLC1A5). A shorter SLC1A5_var that lacked exon 1 of pmSLC1A5, exposing a predicted mitochondrial targeted sequence, was hypothesized to be a candidate mitochondrial glutamine transporter. Mitochondrial localization and glutamine transport activity of SLC1A5_var was confirmed by immunofluorescence co-localization, subcellular fractionation, metabolomics, and stable isotope tracing experiments in cells or isolated mitochondria lacking SLC1A5_var. Notably, SLC1A5_var expression was positively regulated in response to hypoxia (1% O_2_) and hypoxia mimetics (e.g., deferoxamine, cobalt chloride) through a HIF2α-dependent transcriptional mechanism. Although glutaminolysis represents a major carbon source for TCA cycle intermediates in normal conditions, hypoxia and/or mitochondrial dysfunction leads to significant rewiring of glutamine metabolism through reductive carboxylation pathways to support lipogenic flux [214,215,216,217,218]. However, many pancreatic cancer cell lines are capable of sustaining oxidative TCA cycling even at 0.1% O_2_, and Yoo et al. demonstrate that SLC1A5_var activity in pancreatic cancer cells is important for ATP generation from glutamine in hypoxia [77,208]. SLC1A5_var activity promoted glutathione production and ROS scavenging in response to oxygen limitation and was important for gemcitabine resistance mechanisms in cancer cells. Overall, mitochondrial glutamine transport by SLC1A5_var represents an interesting therapeutic strategy for limiting the glutamine demands of cancer cells.

As highlighted above, BCAA catabolism bridges across cytosolic and mitochondrial compartments with transamination and oxidation requiring activity of cytosolic/mitochondrial BCAT1/2 and IMM-localized BCKDH. Recently, Yoneshiro et al. identified that BCAAs serve as important substrates for brown adipose tissue (BAT) metabolism and found that SLC25A44 acts as a key component required for mitochondrial BCAA transport and utilization [210]. Following cold-exposure, plasma levels of valine alone or all three BCAAs decreased in high BAT-containing male adults or obese mice, respectively [210]. ^13^C-labeled leucine contributed significantly to TCA intermediates in human brown adipocytes following noradrenaline treatment, suggesting that mitochondrial oxidation contributes to BCAA clearance in BAT. Furthermore, BAT selectively expresses the mitochondrial BCAT2, not cytosolic BCAT1, thus requiring mitochondrial import. To identify the mitochondrial BCAA transporter, Yoneshiro et al. quantified transcript levels of SLC25 family members and identified several transporters, including uncharacterized SLC25A39 and SLC25A44, of which only SLC25A44 was induced following cold exposure. Functional loss- and gain-of-function experiments and liposomal reconstitution experiments confirmed that SLC25A44 functions as a BCAA transporter required for mitochondrial import required by BAT for thermogenesis and BCAA clearance. BCKAs are also transported across the inner mitochondrial membrane for use as potential acyl-CoA sources. Mitochondrial BCKA transport is facilitated by monocarboxylate transporter 1 (MCT1/SLC16A1), although MCT2/SLC16A2 has also been implicated in certain contexts (e.g., normal brain, breast cancer cell lines) [219,220]. 

In the past few years, significant headway has been made into a more comprehensive understanding of mitochondrial amino acid transporter identity. These studies highlight the diversity of approaches that can be used to identify mitochondrial amino acid transporter function. Cytosolic and mitochondrial amino acid exchange facilitated by mitochondrial transporters is critical for redox shuttle activity (e.g., MAS, FOCM). With the recent identification of key mitochondrial transporters required for amino acid and NAD+ exchange, including SLC25A51 and SLC25A52, we now have the tools necessary to dissect how amino acid redox shuttle activity and/or direct NAD^+^ import influence compartmentalized redox homeostasis [62,63,64,65,66,209,211]. Several amino acid transporters are not yet known, including those for asparagine, tryptophan, alanine, methionine, phenylalanine, tyrosine, cysteine, and proline [206]. While we have known for decades that certain amino acids are metabolized by isolated mitochondria (e.g., proline; Figure 1) [221,222], recent techniques that enable better quantification of mitochondrial metabolism, transport, and metabolite composition will catalyze a deeper understanding of whether mitochondrial transport occurs and which transporters are involved.

## 4. Approaches to Quantify Mitochondrial Metabolism and Transport

Our understanding of metabolite composition within mitochondria and other organelles is mainly derived from our understanding of the metabolic enzymes, transporters, and pathways localized within. Approaches to define the mitochondrial proteome include proteomics analysis of isolated mitochondria, fluorophore-tagging, immunofluorescence, and computational prediction of protein targeting. While reliance of any single approach offers caveats and potential false discovery, cross-referencing of multiple studies provides more accurate prediction of localization. For this reason, MitoMiner (v4.0, 2018) and MitoCarta (v3, 2020) integrate multiple data types and apply machine learning algorithms to provide comprehensive publicly available databases of the mitochondrial proteome [223,224]. Recently, Chen et al. manually curated a list of 346 possible mitochondrial metabolites, referred to as the “MITObolome”, from MitoCarta (v1, 2008) cross-referenced from a list of mitochondrial transporters and enzymes and their substrates extracted from KEGG, which formed the basis for targeted absolute quantification of ~100 metabolites from mitochondria isolated from HeLa cells using a rapid immuno-capture approach [225,226,227]. In contrast, untargeted, “top-down” metabolomic profiling methods have also been used to characterize mitochondrial metabolite composition using traditional differential centrifugation (DC) isolation. For example, Roede et al. used a combination of anion exchange and reverse phase liquid chromatography coupled to mass spectrometry to identify >2100 metabolic features in isolated mitochondria [228]. While there is no consensus of mitochondrial metabolite composition, these studies provide insight into transporter requirements for metabolites not synthesized within the mitochondria. Given the robustness of tagging outward-facing organelle-localized proteins for immuno-capture, several groups have applied this strategy to rapidly isolate lysosomes [229], peroxisomes [230], synaptic vesicles [231], and melanosomes [232] for metabolomic and/or proteomic characterization. Future efforts to rapidly fractionate intracellular compartments whilst preserving metabolite composition will add to our growing understanding of metabolic compartmentalization in relevant contexts and in vivo [233].

Alternative strategies have also been applied to understand mitochondrial metabolic compartmentalization, including selective permeabilization of the plasma membrane. Digitonin selectively permeabilizes the plasma membrane through interaction with cholesterol and pore formation, and other permeabilization agents have also been applied for a similar aim (e.g., saponin, recombinant perfringolysin O) [234,235]. Selective permeabilization has been used to separate mitochondria and cytoplasm for decades [236], and recent efforts to optimize this methodology have resulted in new approaches to measure and/or estimate compartmentalized metabolic flux. Nonnenmacher et al. used ^13^C-labeled pyruvate and glutamine in digitonin-permeabilized A549 cells to quantify how mitochondrial utilization of two major fuels is affected by pharmacological and genetic perturbations [237]. Similarly, but in mitochondria isolated by DC from skeletal muscle and cultured muscle cells, Gravel et al. applied stable-isotope tracing using ^13^C-pyruvate and unlabeled malate to quantify TCA cycle activity in response to pharmacological ETC inhibition [238]. Optimized digitonin permeabilization enabled rapid cytosolic separation from intracellular organelles, that include mitochondria, in as short as 25 s with 90% purity [239]. While sacrificing purity for speed, Lee et al. were able to computationally predict flux distribution and directionality across metabolic pathways localized in both the cytosol and mitochondria (e.g., isocitrate dehydrogenase 1/2/3) [239]. In general, speed and purity are major concerns when isolating mitochondria for downstream metabolite profiling as certain metabolites exhibit high turnover rates that may affect their levels during isolation and post-extraction (e.g., pyruvate, ATP, NADH), convoluting biological interpretation [240,241]. For targeted pathway analysis, inhibitors that prevent enzymatic conversion during purification have successfully been used to characterize lactate metabolism by mitochondrial LDH and may be important to isolate transport activity [242,243].

Notably, many of these approaches can be used to predict and quantify mitochondrial amino acid metabolism and transport. For example, stable-isotope tracing in plasma membrane permeabilized conditions or prior to mitochondrial isolation provides a quantitative means of measuring rates of amino acid transport and catabolism. In addition, high purity mitochondrial isolation and proteomic characterization provides a detailed “menu” of transporters expressed in a particular cellular or environmental context. Classical molecular approaches, including proteoliposome reconstitution, will also continue to be invaluable in characterizing the function and functional regulation of specific transporters [244,245,246,247,248,249,250,251]. 

## 5. Conclusions

Amino acid metabolism is complex and regulated by compartmentalization into distinct subcellular organelles, transporter-mediated exchange, and cellular demands. Despite playing a critical role in regulating metabolic activity, mitochondrial transporters are poorly characterized. Advances in genetic and analytical techniques will shed light into this important class of metabolic regulators.

## Figures and Tables

**Figure 1 metabolites-11-00112-f001:**
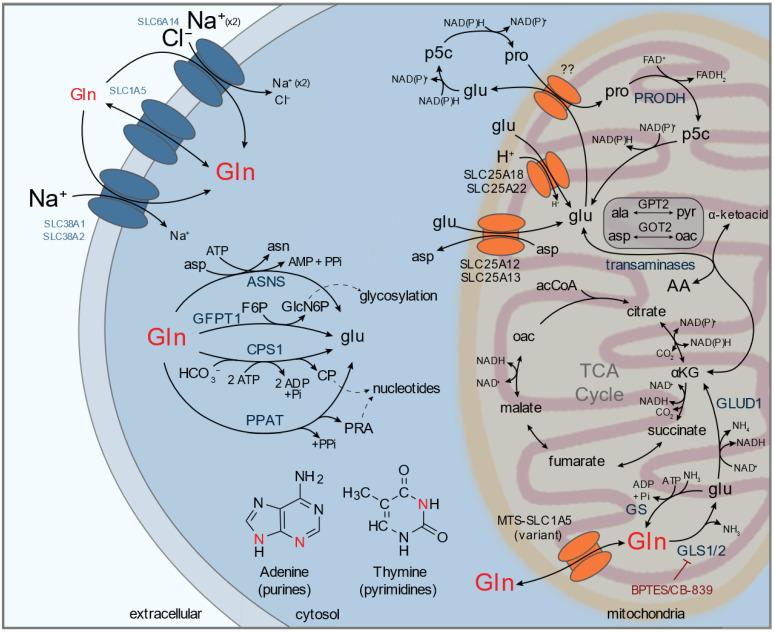
Biochemical pathways and transporters involving glutamine (Gln) and related intermediates. Glutamine is transported by plasma membrane transporters (e.g., SLC1A5/ASCT2, SLC6A14/ATB^0,+^, SLC38A1/SNAT1, SLC38A2/SNAT2) and fuel nucleotide, amino acid, and glycosyl synthesis via asparagine synthetase (ASNS), carbamoyl phosphate synthetase I (CPS1), phosphoribosyl pyrophosphate amidotransferase (PPAT), and glutamine-fructose 6-phosphate aminotransferase (GFPT1). Sodium (Na^+^) and chloride (Cl^−^) gradients across the plasma membrane determine the intracellular concentration of glutamine. Glutamine directly contributes nitrogen for purine and pyrimidine biosynthesis (marked in red). Cytosolic glutamine can also transport into mitochondria via a mitochondrial-targeted (MTS) SLC1A5 variant (MTS-SLC1A5; also referred to as SLC1A5_var) where it acts as a major anaplerotic source for tricarboxylic acid (TCA) cycle metabolism (‘glutaminolysis’). Glutaminolysis is inhibited by BPTES or CB-839, which specifically target glutaminase (GLS). Glutamine-derived glutamate is a significant source of carbon and nitrogen for non-essential amino acid synthesis. αKG, α-ketoglutarate; Ala, alanine; Asp, aspartate; CP, carbamoyl phosphate; F6P, fructose 6-phosphate; GlcN6P, glucosamine 6-phosphate; Gln, glutamine; Glu, glutamate; Oac, oxaloacetate; P5C, pyrroline 5-carboxylate; PRA, 5-phospho-β-d-ribosylamine; Pro, proline; Pyr, pyruvate.

**Figure 2 metabolites-11-00112-f002:**
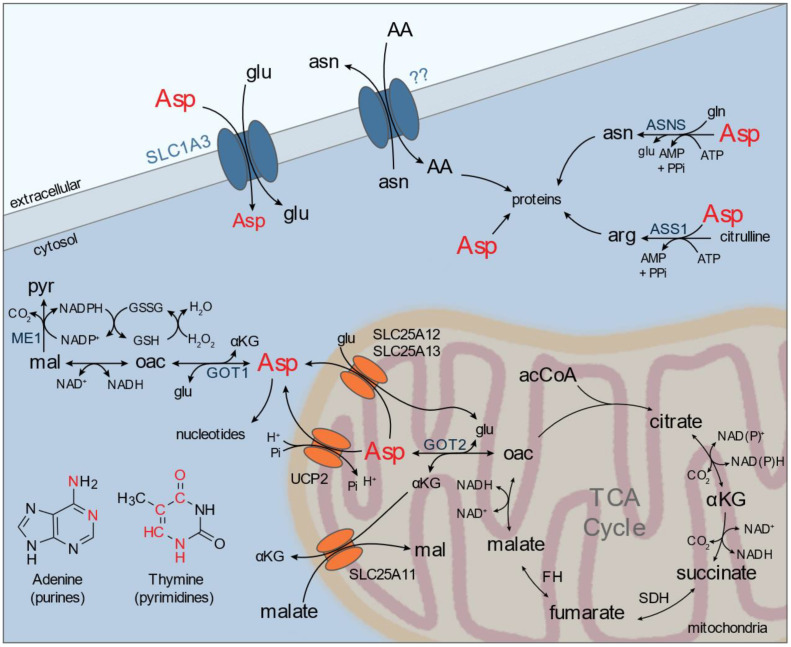
Biochemical pathways and transporters involving aspartate (Asp) and related intermediates. Aspartate is transported by the plasma membrane transporter SLC1A3, which also transports glutamate. Aspartate is synthesized by glutamic-oxaloacetic transaminases (GOT) present in the cytosol (GOT1) or mitochondria (GOT2). Mitochondrial efflux of aspartate mainly occurs through SLC25A12 or SLC25A13, which counter-exchange glutamate and are critical components of the malate-aspartate-shuttle (MAS), and UCP2. Cytosolic aspartate is used as a substrate for asparagine and arginine synthesis via asparagine synthetase (ASNS) and argininosuccinate synthase (ASS1) and as a substrate for nucleotide biosynthesis, contributing carbon and nitrogen to purine and pyrimidines (marked in red). Cytosolic asparagine is used as an exchange factor for several amino acids through an unknown plasma membrane transporter. AA, amino acid; AcCoA, acetyl-coenzyme A; αKG, α-ketoglutarate; Asn, asparagine; Asp, aspartate; FH, fumarate hydratase; Gln, glutamine; Glu, glutamate; GSH, reduced glutathione; GSSG, oxidized glutathione; Mal, malate; Oac, oxaloacetate; Pyr, pyruvate; SDH, succinate dehydrogenase; UCP2, uncoupling protein 2.

**Figure 3 metabolites-11-00112-f003:**
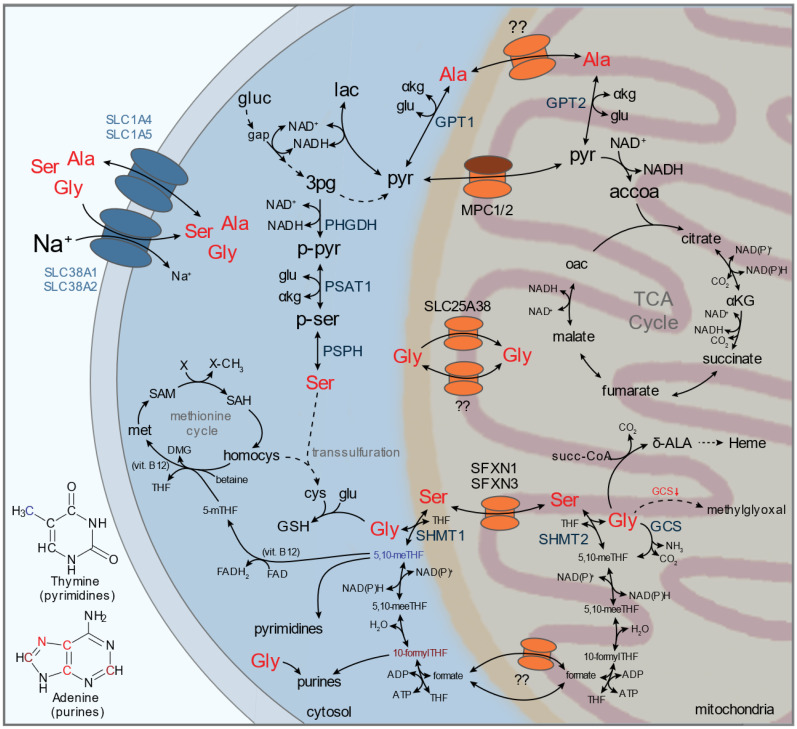
Metabolism of small, neutral amino acids including serine (Ser), glycine (Gly), and alanine (Ala). Serine, glycine, and alanine are mainly transported by the plasma membrane transporters SLC38A1, SLC38A2, SLC1A4, and SLC1A5 or synthesized de novo by cytosolic and/or mitochondrial pathways. Sodium (Na^+^) gradients across the plasma membrane drive intracellular concentration of serine, glycine, and alanine. Serine is synthesized from 3-phosphoglycerate (3pg) through a three-step process involving 3-phosphoglycerate dehydrogenase (PHGDH), phosphoserine aminotransferase (PSAT1), and phosphoserine phosphatase (PSPH). Cytosolic serine is used for several metabolic pathways, including the transsulfuration pathway for de novo cysteine synthesis and folate-mediated one carbon metabolism (FOCM) involving serine hydroxymethyltransferase (SHMT) and methylenetetrahydrofolate dehydrogenases (MTHFD). FOCM occurs in both the cytosol and mitochondria and can produce glycine. In the cytosol, glycine is utilized for glutathione synthesis and purine synthesis (marked in red) and acts as a substrate for the methionine cycle important for methylation reactions. In the mitochondria, glycine can be cleaved by the glycine cleavage system (GCS) to provide one carbon unites for FOCM and is a substrate for δ-aminolevulinic acid (δ-ALA) synthesis. When GCS activity is low, mitochondrial glycine can lead to the accumulation of methylglyoxal. Components of FOCM, including 5,10-methylene-tetrahydrofolate (5,10-meTHF) and 10-formyl-tetrahydrofolate (10-formylTHF) can contribute to nucleotide biosynthesis (marked in dark blue and dark red, respectively). Mitochondrial serine import is facilitated by sideroflexin 1 and 3 (SFXN1/3), formate is transported by an unknown transport(er) mechanism, and glycine is imported by SLC25A38 and other transporters may be involved. Alanine is synthesized from pyruvate by cytosolic or mitochondrial glutamic-pyruvic transaminases (GPT) localized to the cytosol (GPT1) or mitochondria (GPT2). Pyruvate is imported into the mitochondria by the mitochondrial pyruvate carrier (MPC1/2) an obligate heterodimer. Alanine is exchanged between the cytosol and mitochondria by an unknown transporter. 10-formylTHF, 10-formyl-tetrahydrofolate; 3pg, 3-phosphoglycerate; 5,10-meeTHF, 5,10-methenyl-tetrahydrofolate; 5,10-meTHF, 5,10-methylene-tetrahydrofolate; 5-mTHF, 5-methyl-tetrahydrofolate; accoa, acetyl-coenzyme A; αKG, α-ketoglutarate; Ala, alanine; Cys, cysteine; δ-ALA, δ-aminolevulinic acid; DMG, dimethylglycine; Gap, glyceraldehyde 3-phosphate; Gluc, glucose; Gly, glycine; GSH, reduced glutathione; Homocys, homocysteine; Lac, lactate; Met, methionine; Oac, oxaloacetate; p-Pyr, 3-phosphopyruvate; p-Ser, phosphoserine; Pyr, pyruvate; SAH, S-adenosylhomocysteine; SAM, S-adenosylmethionine; Ser, serine; Succ-coa, succinyl-coenzyme A; THF, tetrahydrofolate.

**Figure 4 metabolites-11-00112-f004:**
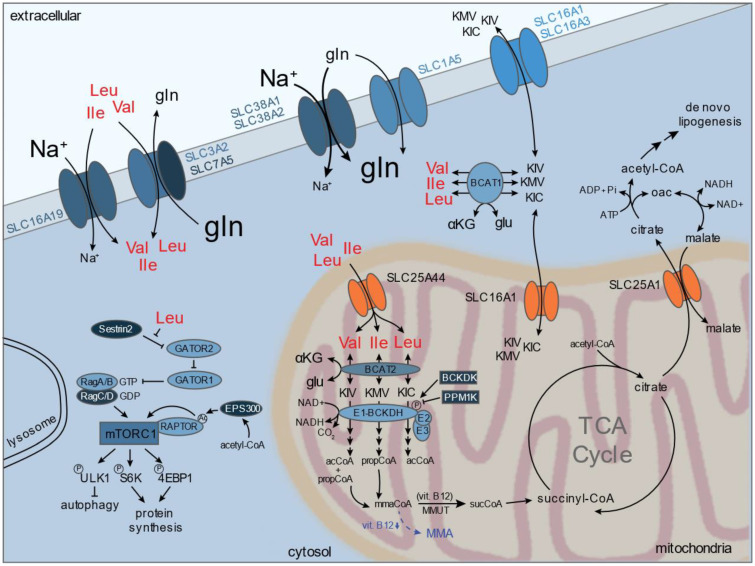
Biochemical pathways and sensing mechanisms involving the branched-chain amino acids (BCAAs) leucine (Leu), isoleucine (Ile), and valine (Val). BCAAs are mainly imported by the large amino acid transporter (LAT1), a dimer consisting of SLC7A5 and SLC3A2/CD98, which functions as an amino acid exchanger and the Na^+^-dependent SLC16A9/B0AT1. Glutamine—mainly transported by SLC38A1, SLC38A2, and SLC1A5—is thought to provide the chemical driving force necessary to influx BCAAs through LAT1. Cytosolic leucine is directly sensed by Sestrin2 and regulates mTORC1-dependent signals that control autophagy, protein synthesis, and proliferation. BCAAs can be metabolized by the BCAA transaminase (BCAT) present in the cytosol (BCAT1) or mitochondrial (BCAT2), which produce branched-chain ketoacids (BCKAs). Cytosolic BCKAs can be transported through SLC16A monocarboxylate transporters present on the plasma membrane or mitochondria. BCAAs are imported into the mitochondria by SLC25A44 and can contribute to acyl-CoA production through activity of BCAT2 and BCKA dehydrogenase (BCKDH), which is regulated by BCKDH kinase (BCKDK) and Mg^2+^/Mn^2+^-dependent 1K protein phosphatase (PPM1K). Acyl-CoA produced by BCAA catabolism can fuel TCA cycle metabolism and de novo lipogenesis. Acetyl-CoA levels are sensed through EPS300-dependent acetylation of RAPTOR, which in-turn regulates mTORC1 activity. Mitochondrial propionyl-CoA, produced either from valine or isoleucine catabolism, is metabolized to produce succinyl-CoA, but can produce the byproduct methylmalonate (MMA) when vitamin B12 levels are low. AcCoA, acetyl-coenzyme A; αKG, α-ketoglutarate; Gln, glutamine; Glu, glutamate; Ile, isoleucine; KIC, α-ketoisocaproic acid; KIV, α-ketoisovaleric; KMV, α-keto-β-methylvaleric; Leu, leucine; Oac, oxaloacetate; PropCoA, propionyl-coenzyme A; SucCoA, succinyl-coenzyme A; Val, valine.

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
