# Peer review of "Transporters at the Interface between Cytosolic and Mitochondrial Amino Acid Metabolism"

_metabolites, 2021, doi:10.3390/metabo11020112_

Round 1

Reviewer 1 Report

This review offers an up to date view on the involvement of transporters in amino acid trafficking between cytosol and mitochondria. Though, The topic is inclusive in major content, well-organized, and easy to read, which shows the significance of a number of amino acids and several regulatory metabolic enzymes in disease pathogenesis. However, authors can highlight and list the recent studies on amino acid transporters, which have significance in translational medicine i.e. therapeutic candidates and key drivers.
Since the mitochondrial protein is significantly affected by intracellular and mitochondrial redox balance (NADP/NADPH etc), the Authors can highlight the pros and cons of redox modifications of amino acids and their transports in cell homeostasis and in pathophysiology of disease .
Though the current article highlighted major AAs and their association in redox balance i.e. NAD+/NADH, NADP+/NADPH, etc. Authors can articulate the significance of NAD+ transporter i.e SLC25A51 and its possible role in the amino acid homeostasis?
Please include a list of abbreviations.

Author Response

We thank the reviewer for their enthusiasm for our manuscript. We have addressed the reviewer's concerns in full (see attached PDF). In the revised manuscript, we have underlined and colored changes in blue. 

Reviewer 2 Report

Comments:

The review

 “Transporters at the Interface Between Cytosolic and Mitochondrial Amino Acid Metabolism”

has been written with the aim to put together and shed light on the scientific knowledge concerning the amino acids transport  both at the plasma membrane and mitochondrial membrane level. The amount of information can overall be useful to get into the “world” of amino acids transport/metabolism, also thanks to appropriate reference use. Indeed, according to my opinion, the reported data/information must be differently organized.

First of all the title:

perhaps the word “interface” is not adequate and does not fits with the content of the review. Second, allover in the paper there are continuous references to the relationship between the amino acids metabolism/transport with cancer and pathologies, so even if this was not the initial “idea” of the review this correlations should be mentioned in the title if the review will maintain the same approach.

The paper:

In lane 38 of the introduction you refer to the correlations with human disease as a specific focus of the review, while the object of the paper, as reported in the title, is written in the subsequent lanes (38-40). So according to the aim of the review in the present form I suggest to invert the sentences.

2.1 glutamine

After the analysis of glutamine transport and metabolism you suddenly, in lane 70, refer to pancreatic cancer etc…to pass to the proline catabolism (lanes 73-74) related glutamate and metastasis from breast cancer. Then you jump to other correlation to glioma cells (lane 76). Then you go again to “normal” metabolic conditions for example writing about neonatal mammal (lane 81) and then again to glutaminolysis in the cancer context (Lanes 81-82). From lane 83 to 131 you also refer to pathologies. From then to the end of the 2.1 paragraph you focus on possible anti cancer drugs. This to point that the paragraph focuses more to the pathological aspects related to glutamine. 

2.2 aspartate

From lane 157 to 188 you explain aspartate metabolism, then from lane 188 you refer to cancer. You explain the influence of NO signal molecule which is sensitized from arginine (lane 218), but arginine is also transported into the mitochondria by Slc25A29 (JBC-Porcelli et al 2014) and this mitochondrial carrier is also involved in cancer modulation (Oncogene-Huiyuan Zhang et al 2018).

2.3 and the other paragraph respect the same scheme, a kind of mix between physiological and pathological involvement of the amino acid metabolism.

Suggestion:

The review puts together a huge amount of information, is well supported by references, but personally I would organize it in a different manner. I would split each paragraph in 2 parts. The first should explain the physiological transport and correlated metabolism, clearly specifying which reactions occur in the cytosol and which occur into the mitochondria. This indeed appears in the figs, but more reference in the text to the Figs can increase the clarity of the arguments. The second part of the paragraph should deal with the correlation with the pathology and since there are a lot of correlations, the addition of a table, in each paragraph, in which are reported

 >> protein: cancer correlated: number of reference

should improve the clarity of the review. This table can be a great effort, and for me it is not mandatory, but if you agree with me and this is feasible I warmly suggest it.

Figures can be more clear using different colors for the different proteins, this can help to understand the complexity of the substrate transport. You have already done this but only for few proteins , for example in Fig 3 MPC1/2 are indicated  with different colors. In the same fig. at a first glance Slc38A1/2 and Slac1A4/5 appear as the same proteins.

In Fig 1 Gln is written in bigger character outside the cell , is it a mistake or it has a meaning? The same in Fig.3 for Na+?

The paragraphs 2.6 gives a huge overview to the approaches used to get deeper inside the mitochondrial metabolism and transport, but indeed omits the molecular approach widely used in the last decades to study great part of the mitochondrial Slc25 carriers based on their reconstitution in liposomes, carriers either extracted from tissues or recombinant. You cited 2 reviews in the previous paragraph (ref 184-185) these excellent reviews are indeed focused on disease and  pathology. There are also paper in which the molecular approach lead to demonstrate the channeling of substrate and proteins at the mitochondrial level (Console- Mol Cell Biochem-2014), as well as several signals and post transalational modifications acting on carriers which can tune their transport activity and indeed the mitochondrial and cellular metabolism ( Giangregorio- Mol Cell Biochem-2017) (Tonazzi- BBA – Bioenergetics-2017).

Minor revision

In lane 240 I argue that you meant not “synthetic enzymes” but: enzymes involved in the synthesis …

Lane 262 : formate which can transport, should be: formate which can be transported

Author Response

We thank the reviewer for their critical feedback on our review. We have incorporated the reviewer's comments into our revised manuscript. Please see attached a detailed description of the changes made. In the revised manuscript, we have indicated text changes as underlined and blue colored.  

Reviewer 3 Report

Amino acid metabolism is an actual issue and a very important and complex biochemical process in the human body. In their review, Keeley G et al. elucidate key metabolic outputs of eukaryotic mitochondrial amino acid metabolism and also discuss membrane and mitochondrial transporters, which act as important interfaces between the cellular compartments.

This review has some merit because summaries of this complex issue are rare in the current literature. The authors provide a good overview also about the special metabolism of the branched chain amino acids. They present the manuscript in a good scientific English.

To my opinion this article can be published in “Metabolites” in the current form.

Author Response

We thank the reviewer for their enthusiasm for our review. We have incorporated changes into the revised version to include suggestions from the other reviewers. These changes can be found as blue, underlined text in the revised manuscript. 

Round 2

Reviewer 2 Report

Lane 14: to avoid reduntant "discuss" in lanes 13 and 14 I suggest to substitute the one in line 14 with "analyze", and in the same sentence I should write "metabolism functions in physiological and disease. contexts".

In lane 17 remove the comma before and.

lane 187 .:TRANSPORTERS should be transports